# Cross-Domain Adaptive Transfer Reinforcement Learning Based on State-Action Correspondence

**Heng You**[1]     **Tianpei Yang** [*1,2]     **Yan Zheng**[1]     **Jianye Hao** [*1]     **Matthew E. Taylor**[2]

[1]College of Intelligence and Computing, Tianjin University, China
[2]Department of Computing Science, University of Alberta and Alberta Machine Intelligence Institute, Canada

## Abstract

Despite the impressive success achieved in various domains, deep reinforcement learning (DRL) is still faced with the sample inefficiency problem. Transfer learning (TL), which leverages prior knowledge from different but related tasks to accelerate the target task learning, has emerged as a promising direction to improve RL efficiency. The majority of prior work considers TL across tasks with the same state-action spaces, while transferring across domains with different state-action spaces is relatively unexplored. Furthermore, such existing cross-domain transfer approaches only enable transfer from a single source policy, leaving open the important question of how to best transfer from multiple source policies. This paper proposes a novel framework called *Cross-domain Adaptive Transfer* (CAT) to accelerate DRL. CAT learns the state-action correspondence from each source task to the target task and adaptively transfers knowledge from multiple source task policies to the target policy. CAT can be easily combined with existing DRL algorithms and experimental results show that CAT significantly accelerates learning and outperforms other cross-domain transfer methods on multiple continuous action control tasks. The code for this project are released, under the project page of https://github.com/TJU-DRL-LAB/transfer-and-multi-task-reinforcement-learning.

## 1 INTRODUCTION

Deep reinforcement learning (DRL) combining deep neural networks with RL algorithms [Sutton and Barto, 1998] has achieved impressive success in multiple domains like game playing [Mnih et al., 2015, Silver et al., 2016] and continuous control [Lillicrap et al., 2016]. However, DRL is still faced with sample inefficiency problem that requires large amounts of interactions with the environment. Transfer learning (TL), as a technique to accelerate the learning process of RL by leveraging prior knowledge, has become one popular research direction to significantly reduce sample complexity [Taylor and Stone, 2009, Zhu et al., 2020, Yang et al., 2020a, 2021].

One major branch of transfer in RL focuses on leveraging external knowledge from pre-trained policies on source tasks, which we call *policy transfer*. These approaches either distill knowledge from source policies by imitation learning [Rusu et al., 2016, Schmitt et al., 2018, Parisotto et al., 2016, Yang et al., 2020a,b, Tao et al., 2021], or reuse source policies for exploration based on the evaluation of source policies on the target environment [Fernández and Veloso, 2006, Li and Zhang, 2018]. However, all these methods require the same assumption that source tasks share the same state-action space with the target task so that the source policies can be directly imitated or reused. Previous works transfer knowledge between tasks with different state-action spaces based on the hand-coded or learnt mapping [Taylor et al., 2007, 2008] in tabular settings, which can not be applied to more complex tasks. In practice, tasks in real-world scenarios may exhibit many differences, not only in the dynamics and rewards but also in the mismatch between the state-action space. An ideal method should be capable of handling such a mismatch and realize more generalized transfer learning.

Recently, a few approaches consider how to deal with the mismatch in the state-action space by mapping state spaces into a common feature space using a state encoder [Gupta et al., 2017, Wan et al., 2020]. However, these methods suffer either of the following limitations, e.g., Gupta et al. [2017] requires paired data of two tasks collected by pre-trained policies or human labeling to train the encoder, which is a strong assumption and usually expensive for real-world problems. MIKT [Wan et al., 2020] only considers the relevance of the state embedding and the target

---

*Correspondence to Tianpei Yang <tpyang@tju.edu.cn>, Jianye Hao <jianye.hao@tju.edu.cn>

*Accepted for the 38th Conference on Uncertainty in Artificial Intelligence* (UAI 2022).

state to train the encoder, which makes the state embeddings unable to reflect the correlation with the source state, and finally influences the transfer performance. Some works [Chen et al., 2019, Zhang et al., 2021] focus on learning both the mapping between the state spaces and action spaces for transfer. However, the former method can only deal with discrete action spaces. The latter method adopts zero-shot transfer through the mapping, which can not achieve optimal performance on the target task. Furthermore, all above methods only consider learning a one-to-one mapping and transferring a single source policy. This paper instead tackles the more difficult case of learning to transfer from multiple tasks with different state-action spaces.

To this end, we propose a novel framework called *Cross-domain Adaptive Transfer* (CAT), which adaptively transfers multiple source policies with different state-action spaces. Different from previous works, we do not require paired data to learn the state-action correspondence or learn insufficiently trained state correspondence. Instead, CAT learns the state-action correspondence from each source domain to the target domain through a state encoder, action encoder, and reverse state encoder using the trajectories of source policies. Since the source environment is inaccessible for more information, we do not need reverse action encoders to get the actions on the source environment. Besides, CAT learns the state embeddings which can satisfy the properties proposed in Section 3.2 to achieve better transfer performance by proposing extra optimization objectives. Further, CAT evaluates each source policy on the target task and learns how helpful each source policy is to the target policy, and then uses the performance as the measurement to determine when and which source policies should be transferred. In this way, CAT can adaptively transfer multiple cross-domain policies into the target policy. In summary, our contributions are as follows:

- Our novel transfer framework, CAT, consists of three main components: an agent module, a self-adaptive module, and a correction module, to solve the problem of adaptive knowledge transfer from multiple source policies with different state-action spaces.

- CAT learns more sufficiently trained state embeddings and action embeddings using the correction module and the agent module, which serves as the basis of the following transfer process.

- CAT combines knowledge from source policy networks with the target policy network using an adaptive weighting factor generated by the self-adaptation module.

- CAT can be easily combined with existing DRL algorithms and experimental results show that CAT efficiently accelerates RL and outperforms other related transfer methods on continuous control tasks with different state-action spaces.

## 2 BACKGROUND

This section introduces notation and defines our problem setting. We typically model RL problems with a Markov decision process (MDP), which can be described as a tuple $\mathcal{M} = \langle \mathcal{S}, \mathcal{A}, \mathcal{R}, \mathcal{T}, \gamma \rangle$, where $\mathcal{S}$ and $\mathcal{A}$ are the sets of states and actions, respectively; $\mathcal{T} : \mathcal{S} \times \mathcal{A} \times \mathcal{S} \mapsto [0, 1]$ is the state transition probability function; $\mathcal{R} : \mathcal{S} \times \mathcal{A} \times \mathcal{S} \mapsto \mathbb{R}$ is the reward function which gives returns on the agent's performance; and $\gamma$ is the discount factor for future rewards. A policy $\pi : \mathcal{S} \times \mathcal{A} \mapsto [0, 1]$ is defined as a state-conditioned probability distribution over actions and the objective of the agent is to find an optimal policy $\pi^*$ maximizing the expected discounted return $R = \sum_{i=t}^{T} \gamma^{i-t} r_i$.

**Policy Gradient (PG) Algorithms.** Policy gradient methods are widely used to directly optimize the policy $\pi$ parameterized by $\theta$. Proximal policy optimization (PPO, Schulman et al. [2017]) is currently one of the most efficient PG methods. In each iteration, PPO tries to calculate a new policy $\pi_\theta$ and ensure that the difference between $\pi_\theta$ and the rollout policy $\pi_{\theta_{old}}$ is not too large by adding a constraint during the training process. The following loss is minimized over multiple epochs:

$$L_{\text{PPO}}^{\theta} = -\mathbb{E}_\tau \left[ \min \left( r_t(\theta) \hat{A}_t, \text{clip}(r_t(\theta), 1 - \varepsilon, 1 + \varepsilon) \hat{A}_t \right) \right]$$

where $r_t(\theta) = \frac{\pi_\theta(a_t|s_t)}{\pi_{\theta_{old}}(a_t|s_t)}$ is the ratio of the action probabilities under the rollout policy and current policy and $\hat{A}_t$ is the estimated advantage. The value network $V_\psi$ is updated with temporal difference learning: $L_{\text{PPO}}^{\psi} = -\mathbb{E}_\tau[(V_\psi(s_t) - V_t^{\text{targ}})^2]$. The overall PPO minimization objective is:

$$L_{\text{PPO}}(\theta, \psi) = L_{\text{PPO}}^{\theta} + L_{\text{PPO}}^{\psi} \quad (1)$$

**Problem Settings.** Same-domain transfer learning considers a source MDP $\mathcal{M}_{\text{source}} = \langle \mathcal{S}, \mathcal{A}, \mathcal{R}_{\text{source}}, \mathcal{T}_{\text{source}}, \gamma \rangle$ and a target MDP $\mathcal{M}_{\text{target}} = \langle \mathcal{S}, \mathcal{A}, \mathcal{R}_{\text{target}}, \mathcal{T}_{\text{target}}, \gamma \rangle$, where the two MDPs have the same state and action spaces, but other properties such as $\mathcal{R}$ or $\mathcal{T}$ may be different. A standard objective is to accelerate learning the target task by leveraging $\mathcal{M}_{\text{source}}$ (relative to learning from scratch).

In cross-domain transfer, the state and action spaces can be different: $\mathcal{M}_{\text{source}} = \langle \mathcal{S}_{\text{source}}, \mathcal{A}_{\text{source}}, \mathcal{R}_{\text{source}}, \mathcal{T}_{\text{source}}, \gamma_{\text{source}} \rangle$ and $\mathcal{M}_{\text{target}} = \langle \mathcal{S}_{\text{target}}, \mathcal{A}_{\text{target}}, \mathcal{R}_{\text{target}}, \mathcal{T}_{\text{target}}, \gamma_{\text{target}} \rangle$. However, current TL methods are only able to successfully distill knowledge from a single source policy. This paper considers the problem of cross-domain transfer between multiple source tasks and a target task. We denote this as a series of source MDPs $\Pi_{\mathcal{M}} = \{ \mathcal{M}_1, \mathcal{M}_2, \cdots, \mathcal{M}_n \}$ and a target MDP $\mathcal{M}_{\text{target}}$ where $\mathcal{M}_i$ represents the $i$-th source MDP for convenience. We generally assume there are some high-level commonalities between the MDPs (e.g., a quadruped, hexapod, and octopod robots may have qualitatively similar gaits). In this work, our objective is to adaptively transfer knowledge from $\Pi_{\mathcal{M}}$ to accelerate the learning process.

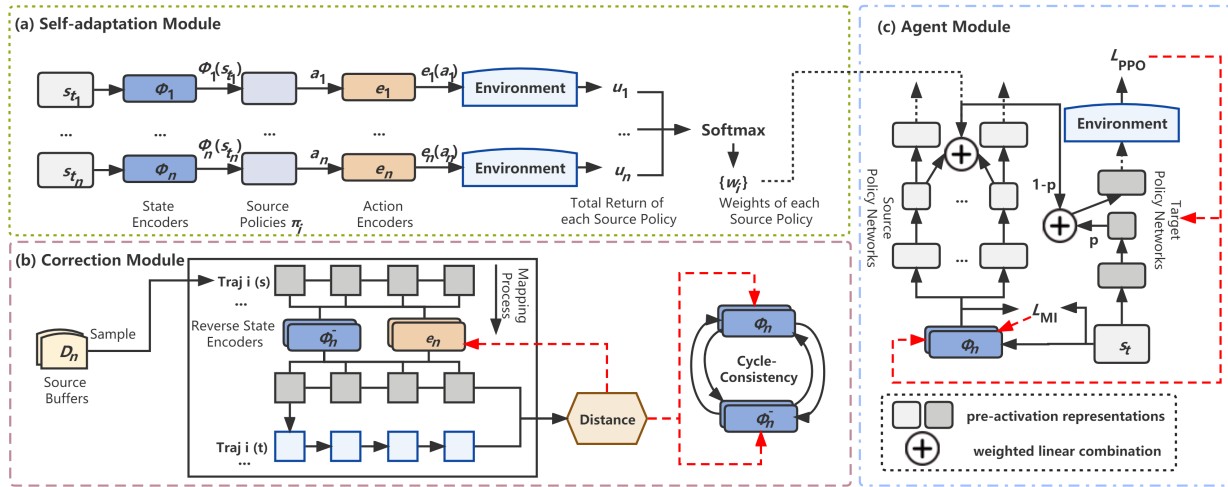

Figure 1: An illustration of our Cross-domain Adaptive Transfer framework which contains three main components: (a) Self-adaptation Module. (b) Correction Module. (c) Agent Module. The state encoders $\phi_n$ are updated using $\{L_{\text{PPO}}, L_{\text{MI}}, L_{\text{cyc}}, L_{\text{corr}}\}$. The reverse state encoders $\phi_n^-$ are updated using $\{L_{\text{cyc}}, L_{\text{corr}}\}$. The action encoders $e_n$ are updated using $\{L_{\text{corr}}\}$. Note that we only show the target policy network, and the value network is the same.

# 3 METHODOLOGY

In this section, we first introduce our whole framework and each component. Then, we describe how to learn state and action embeddings and how to adaptively transfer multiple cross-domain source policies to the target task. Finally, we describe CAT combining with a specific DRL algorithm, PPO [Schulman et al., 2017] in detail.

## 3.1 FRAMEWORK OVERVIEW

Figure 1 illustrates the proposed Cross-domain Adaptive Transfer framework (CAT) which contains three components. The three components described are not completely novel. However, we integrate them into CAT to better solve the cross-domain transfer problem and our empirical ablation studies validate their effectiveness and importance (see Section 4.2).

**Correction Module** Instead of only considering the relevance between state embedding and the target state in MIKT [Wan et al., 2020], we propose four properties that the learned state embedding should satisfy and build two extra optimization objectives to learn state and action embeddings in the correction module, described in Section 3.2. A newly proposed *correction module* (see Figure 1(b)) is used to learn state embeddings that can satisfy these properties and to learn action embeddings that can better capture the semantics of actions of the source and target tasks, both of which are described in Section 3.2. The goal of the correction module is to learn embeddings to distill knowledge from multiple source policies into the target task.

**Self-Adaptation Module** Inspired by existing same-domain transfer methods (e.g., Fernández and Veloso [2006]), we want our method to decide when and which source policy is better to transfer by evaluating them on the target environment. The *self-adaptation module* (see Figure 1(a)) evaluates the source policies (via the relevant embeddings) for a fixed number of steps in the target environment. The average performance lets us set weighting factors so that we can combine these different source policies. We explain this idea in the context of CAT in Section 3.3.

**Agent Module** Once the embeddings are trained (via the correction module) and the source task policies are weighted (via the self-adaptation module), the agent is now ready to learn from both environmental interaction and guidance from the transferred policies. The *agent module* allows our agent to distill knowledge from source policies, select actions to execute in the target environment, and learn a high-performing policy. See Figure 1(c) and Section 3.3.

## 3.2 LEARNING STATE-ACTION CORRESPONDENCE

This section considers how to learn meaningful state and action correspondences. We first introduce a set of state embedding spaces parameterized by a set of encoder functions $\{\phi_1, \phi_2, \cdots, \phi_n\}$. Each state embedding is defined as $\mathcal{S}_{\text{emb}_i} := \{\phi_i(s)|s \in \mathcal{S}_{\text{target}}, \phi_i(s) \in \mathcal{S}_{\text{source}_i}\}$, which will be used to map useful knowledge from source policies into the target policy.

In order to extract more useful knowledge, each state em-

bedding $\mathcal{S}_{\text{emb}_i}$ should satisfy the following four properties: (1) The embeddings should be task-aligned to maximize the cumulative discount rewards in the target MDP. (2) The input states and state embeddings should be highly correlated so that the agent can receive the most appropriate guidance from source policies in the current state. (3) The embeddings should preserve enough information about the source task so that $\phi_i(s)$ can be reconstructed to the target task as consistently as possible. (4) In addition to the correspondence on the single state, $s_s$ and $s_t$, the state embedding should keep the correspondence between state sequences of the source and target tasks.

To achieve property (1), we use the policy gradient to update the state encoder parameters [Wan et al., 2020, Chen et al., 2019]. Property (2) can be achieved by maximizing the mutual information between states and embeddings to achieve a high correlation as follows [Wan et al., 2020]:

$$
\begin{aligned}
\mathcal{I}(s; e) &= \mathcal{H}(s) - \mathcal{H}(s|\phi(s)) \\
&= \mathcal{H}(s) + \mathbb{E}_{s,e}[\log p(s|e)] \\
&= \mathcal{H}(s) + \mathbb{E}_{s,e}[\log q_\omega(s|e)] \\
&\quad + \mathbb{E}_e\left[D_{\text{KL}}(p(s|e)||q_\omega(s|e))\right] \\
&\geq \mathcal{H}(s) + \mathbb{E}_{s,e}[\log q_\omega(s|e)]
\end{aligned}
$$

where $\mathcal{H}$ denotes the differential entropy. The above optimization goal is known as a variational information maximization algorithm and the variational distribution $q_\omega(s|e)$ approximates the true conditional distribution $p(s|e)$. So the final optimization goal can be written as:

$$
L_{\text{MI}}(\phi) = -\mathbb{E}_{s \sim \rho_s}\left[\log q_\omega\big(s|\phi(s)\big)\right] \tag{2}
$$

where $\rho_s$ denotes the state distribution of the target policy.

However, relying only on the above two properties does not guarantee good enough transfer performance of the state embeddings. Property (3) is also applied in Gupta et al. [2017], and Chen et al. [2019] only satisfies property (4) by keeping the correspondence on the single state. Instead, we argue that state embeddings that satisfy all four properties will achieve better transfer performance — this is empirically verified in our experiments.

To this end, in addition to using policy gradient and mutual information to train state embeddings, we propose the **correction module** (Figure 1(b)) to satisfy the remaining two properties. We introduce a set of reverse state embeddings parameterized by a set of decoder functions $\{\phi_1^-, \phi_2^-, \cdots, \phi_n^-\}$ and each reverse state embedding is defined as $\mathcal{S}_{\text{emb}_i}^- := \{\phi_i^-(s)|s \in \mathcal{S}_{\text{source}}, \phi_i^-(s) \in \mathcal{S}_{\text{target}}\}$.

In our method, we use the reverse state embeddings in two ways to build two types of optimization objectives corresponding to properties (3) and (4), respectively. Firstly, a pair of meaningful mapping functions $\phi$ and $\phi^-$ should be as invertible as possible: $\phi^-\big(\phi(s_t)\big) \approx s_t$, $\phi\big(\phi^-(s_s)\big) \approx s_s$,

so that the state embeddings $\mathcal{S}_{\text{emb}}$ can preserve as much information about the source domain as possible [Gupta et al., 2017]. Therefore, it is expected that state embeddings $\mathcal{S}_{\text{emb}}$ can map from the embedding spaces back to their original state spaces. To satisfy property (3), we define the *cycle-consistency loss* as follows:

$$
\begin{aligned}
L_{\text{cyc}}(\phi, \phi^-) &= \mathbb{E}_{s_t}\left[||\phi^-\big(\phi(s_t)\big) - s_t||_2\right] \\
&\quad + \mathbb{E}_{s_s \sim \tau_s}\left[||\phi\big(\phi^-(s_s)\big) - s_s||_2\right]
\end{aligned} \tag{3}
$$

where $\tau_s$ denotes the trajectories of the source policies. Secondly, we minimize the deviation between the mapping state sequence and the real state sequence to satisfy the property (4). Specifically, trajectories $\langle s_{s_T}, a_{s_T}, s_{s_{T+1}} \rangle$ sampled from source buffers which are collected during the training of source policies are mapped to the target environment $\langle \phi^-(s_{s_T}), e(a_{s_T}), \phi^-(s_{s_{T+1}})) \rangle$ through the reverse state encoders and action encoders. Then, given the initial state $s_{s_0}$ of each source trajectory, we can obtain a true trajectory $\langle \langle \phi^-(s_{s_0}), e(a_{s_0}), s_{t_1}) \rangle, \ldots, \langle s_{t_T}, e(a_{s_T}), s_{t_{T+1}}) \rangle \rangle$, starting with the mapped initial state $\phi^-(s_{s_0})$, by interacting with the target environment using each mapped action $e(a_{s_T})$ at each following state. Figure 2 shows the first step derivation calculation process. To satisfy property (4), the *correction loss* calculates the total derivation over trajectories as follows:

$$
L_{\text{corr}}(\phi, \phi^-, e) = \mathbb{E}_{s_s \sim \tau_s, s_t \sim \tau_t}\left[||\phi^-(s_{s_{T+1}}) - s_{t_{T+1}}||_2^2\right] \tag{4}
$$

where $\tau_s$ and $\tau_t$ denote the sampled trajectory and the true trajectory, respectively. Note that it is not necessary for the reverse state embeddings to satisfy properties (1) and (2) since they just need to make sure the successful reconstruction of state embeddings and keep the correspondence between state sequences. Therefore, the reverse state encoders are updated only using the cycle-consistency loss and the correction loss in the correction module.

Next, we introduce how to learn meaningful action correspondence which can capture the semantics of actions in the correction module. We introduce a set of action embedding spaces parameterized by a set of encoder functions $\{e_1, e_2, \cdots, e_n\}$. We follow the main idea that the semantics of actions can be reflected by their effects on the environment, which can be measured by the state transition probability in RL [Chen et al., 2019]. The action embeddings can satisfy the property that the distance should be adjacent if the actions have similar effects by minimizing their effects on the environment, which can be reflected in Equation 4. Therefore, the action encoders are updated using the correction loss in the correction module.

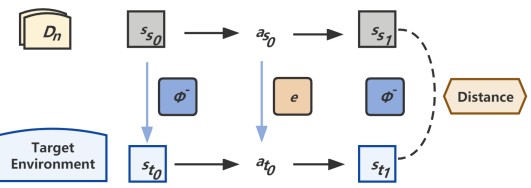

Figure 2: The first step of computing the distance of the obtained trajectory and the real trajectory.

## 3.3 ADAPTIVE POLICY TRANSFER

In this section, we describe how to transfer knowledge adaptively from multiple source policies with learned state-action correspondence through the correction module. The first issue is how to determine when and which source policy should be transferred to the target task. This is achieved through the **self-adaptation module**, which evaluates the source policies and generates the weights for transferring different source task policies (Figure 1(a)). Specifically, the self-adaptation module first evaluates each source policy's performance on the target environment and uses the total return $u_i$ on a fixed number of episodes as the weight of each source policy in the next iteration after passing through the softmax function:

$$w_i = \frac{\exp(u_i)}{\sum_{n=1}^{N} \exp(u_n)}$$

This approach is common but the most intuitive and interpretive way to measure each source policy. With the weighting factor generated by the self-adaptation module, the **agent module** (Figure 1(c)) makes decisions by adaptively drawing out suitable knowledge from multiple source policies and value networks, denoted as $\pi_{\theta'_i}$ and $V_{\psi'_i}$ respectively. In general, we assume the source and target policy and value networks to have the same number of hidden layers $N_\pi$ for ease of exposition. Specifically, the agent module serves the current state $s_t$ and feeds it to the set of encoders to produce state embeddings $\mathcal{S}_{\text{emb}}$, which can be readily passed through the source networks to extract $\{z_{\theta'_i}^j, z_{\psi'_i}^j, 1 \le j \le N_\pi, 1 \le i \le N\}$, representing the pre-activation outputs of the $j$-th hidden layers of the $i$-th source policy and value networks. To get the pre-activation representations $\{z_{\pi_\theta}^j, z_{V_\psi}^j\}$ in the target networks, we used two weighted linear combinations, one for the outputs from the source and target networks and the other for outputs from multiple source policy networks:

$$
\begin{aligned}
z_{\pi_\theta^j} &= p z_\theta^j + (1-p) \sum_{i=1}^{N} w_i z_{\theta'_i}^j \\
z_{V_\psi^j} &= p z_\psi^j + (1-p) \sum_{i=1}^{N} w_i z_{\psi'_i}^j
\end{aligned}
\tag{5}
$$

where $w_i$ is the weight of source policy $\pi_i$. $p \in [0,1]$ is an

increasing factor over time that controls the decrease of the influence of source policies on the target policy — A higher value of $p$ means the lesser influence. Besides, a higher value of $w_i$ means the average performance of the corresponding source policy on the target task is higher. Such a source policy can provide more beneficial knowledge. Meanwhile, at the beginning of the training, the agent selects an action relying more on source policies to gain assistance. As the training continues, the agent should focus more on the target task to avoid negative transfer. In this way, CAT more fully combines knowledge from multiple source task policies to facilitate more efficient learning.

## 3.4 CAT-PPO

This section details CAT-PPO, where we integrate PPO [Schulman et al., 2017] into our framework. As shown in Algorithm 1, other DRL algorithms could easily be incorporated instead. CAT-PPO first initializes all the network parameters needed in the learning process (Line 1). In each iteration, the correction module first samples trajectories from each source task buffer to train all encoders following Section 3.2 (Lines 4-6). Then, the self-adaptation module evaluates each source policy and gets the corresponding weight through state and action embeddings (Line 8). Next, the agent module outputs actions by combining knowledge from the target policy network and source policy networks (see Section 3.3). These actions are executed in the environment to collect trajectories (Line 10). Finally, the agent module computes the RL loss (Equation 1) and the mutual information loss (Equation 2) for the update (Lines 11-13).

---

**Algorithm 1:** CAT-PPO

---

1 **Initialize:** state encoder parameters $\phi_i$, reverse state encoder parameters $\phi_i^-$, action encoder parameters $e_i$, target policy and value network parameters $\theta, \psi$, source buffer $\mathcal{D}_i$

2 **repeat**

3    // Correction module

4    Sample a batch of trajectories from each $\mathcal{D}_i$

5    Update each $\phi_i, \phi_i^-$         $\triangleright$ see Eq. (3)

6    Update each $\phi_i, \phi_i^-, e_i$       $\triangleright$ see Eq. (4)

7    // Self-adaptation module

8    Evaluate each source policy $\pi_i$ and calculate $w_i$

9    // Agent module

10   Collect trajectories $\tau$ by combining the target policy network and source policy networks using $w_i$   $\triangleright$ see Eq. (5)

11   **for** *each batch* $m \in \tau$ **do**

12      Update $\theta, \psi$ with $\nabla_{\theta,\psi} L_{\text{PPO}}(\theta, \psi, \phi)$

13      Update each $\phi_i$ with $\nabla_\phi L_{\text{MI}}(\phi) + \nabla_\phi L_{\text{PPO}}(\theta, \psi, \phi)$

14 **until** *reaching maximum training steps*;

---

# 4 EXPERIMENTS

In this section, we conduct extensive experiments to verify the effectiveness of our proposed algorithm compared with previous cross-domain transfer methods. Further, we design several ablation studies to analyze the contribution of each proposed module to the transfer performance. We also test the influence of different transfer manners in CAT on the final performance to validate the choice in this paper, which is detailed in the Appendix. Results are averaged over 5 different random seeds and each seed with 2 million timesteps of environment interactions. Please see Appendix for the network structure and parameter settings used in this paper.

**Environments**: We use a series of environments provided by Wang et al. [2018], which have a similar physical structure to a centipede. In these environments, the **Centipede** agent consists of repetitive torso bodies, each of which has two legs, and needs to learn to run in a particular direction. The agent is rewarded for running speed and whether it runs within a valid range and penalized for energy cost and resistance obstruction from the ground. To make the experiment more convincing, we consider the **CrippleCentipede** agent, which has two back legs disabled and we denote it as **CpCentipede**. In addition, we also considered the standard **Ant-v2** task from the MuJoCo suite. Figure 3 shows an illustration of all the different scenarios mentioned above. All the source policies are obtained by learning from scratch using the standard DRL method PPO. Please see Appendix for a more detailed description.

**Baselines**: Because the cross-domain transfer methods mentioned in Section 1 suffer various limitations and can not tackle our more difficult setting, we consider the following three baselines:

- Standard DRL method PPO [Schulman et al., 2017], which learns from scratch in the target task;

- MIKT [Wan et al., 2020], which realizes cross-domain knowledge transfer with a single source policy;

- MIKT-MULTI, which is an extended version of MIKT to the setting of multiple source policies. MIKT-MULTI uses a fixed weighting factor to extract knowledge from each source policy, which can be seen as a version of CAT without the self-adaptation module and correction module.

## 4.1 EXPERIMENTAL RESULTS

In our experiments, we design six different combinations of environments to extensively validate the efficiency of our proposed method. For example, "4,6-8" represents **CentipedeFour** and **CentipedeSix** transfer to **CentipedeEight** in Figure 4(a). All the source policies are trained from scratch on source tasks. We plot the average episodic returns on the y-axis (mean and standard deviation). In each

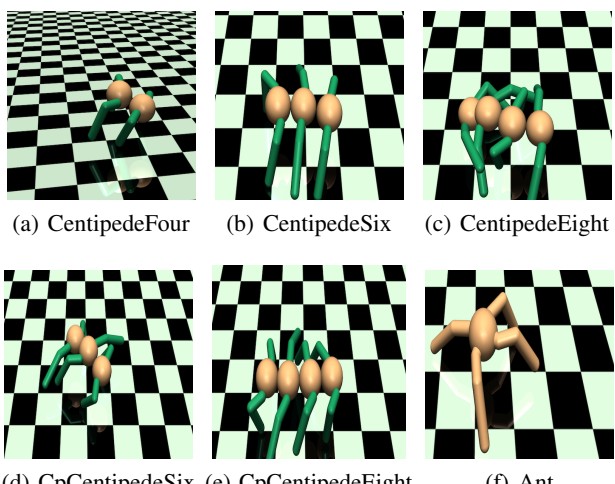

(a) CentipedeFour    (b) CentipedeSix    (c) CentipedeEight

(d) CpCentipedeSix   (e) CpCentipedeEight    (f) Ant

Figure 3: Our continuous control tasks on MuJoCo: Centipede-{4,6,8}, CpCentipede-{6,8} and Ant-v2.

plot, we only consider the source policy which can achieve better transfer performance of MIKT since MIKT can only transfer a single source policy. For example, only "6-8" is plotted in Figure 4(a) since the source policy from **CentipedeSix** provides better transfer performance.

Figure 4 shows the performance of CAT and the other three baselines in different combinations of environments. We can see that the performance of PPO learning from scratch is the worst because of the sample inefficiency and the lack of the help of expert knowledge. Although MIKT achieves better performance than PPO, it is worse than CAT in terms of learning speed and final performance. MIKT-MULTI performs even worse than MIKT in most cases. This indicates that using fixed (or manually adjusted) transfer weights among multiple source policies limits access to more beneficial knowledge or even causes negative transfer, which is exactly what our proposed method aims to solve. This phenomenon further validates the importance of our self-adaptation module and correction module. Finally, we can see that our method (CAT) significantly outperforms all baselines and achieves the highest average rewards with the fastest speed. This is because CAT learns more sufficiently trained state correspondence by satisfying our proposed properties at the same time to lay the foundation for our transfer framework. In addition, it effectively leverages the evaluation performance as the weights of different source policies in the target environment so that it can infer when and which source policy is more beneficial to achieve adaptive knowledge transfer.

**Other domains and more than two source tasks**: In addition to **Centipede-x**, we have some other series of environments, such as **InvPendulum-x**, **Reacher-x**, and **Snake-x**, where $x$ represents the number of joints. In each series of environments, the robotics share an inherent structure that could be exploited for transfer learning. In these environ-

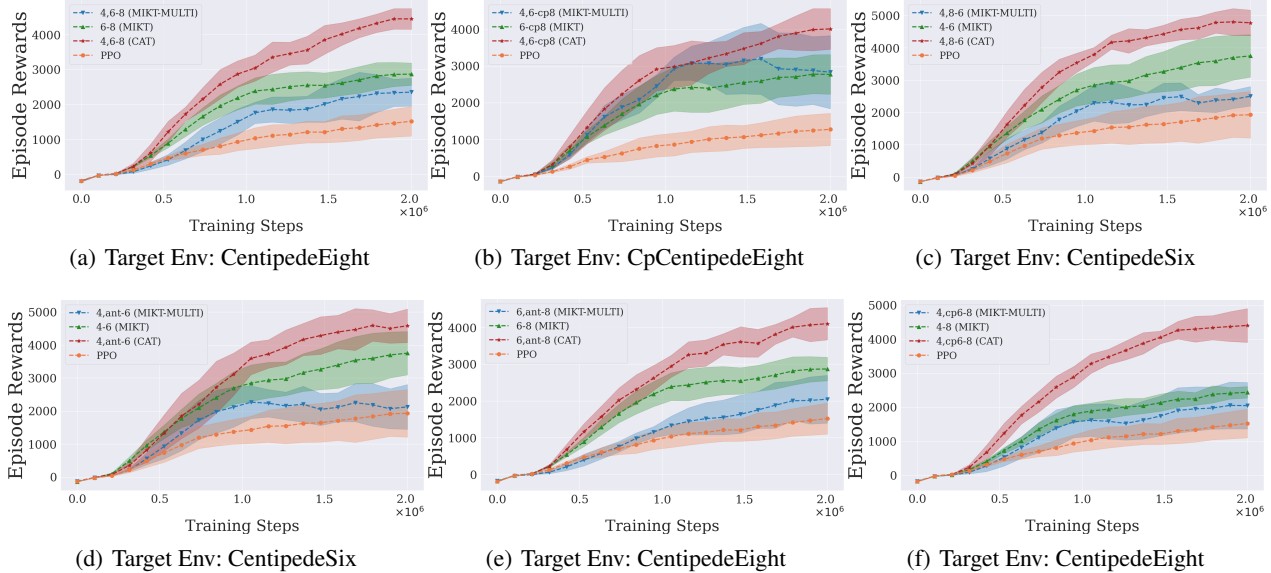

(a) Target Env: CentipedeEight    (b) Target Env: CpCentipedeEight    (c) Target Env: CentipedeSix

(d) Target Env: CentipedeSix    (e) Target Env: CentipedeEight    (f) Target Env: CentipedeEight

Figure 4: Performance of our proposed algorithm (CAT) and other methods (PPO, MIKT, and MIKT-MULTI) on different combinations of continuous control tasks. We plot the number of timesteps of environment interaction on the x-axis and the average episodic returns on the y-axis (the curves and shadow areas represent the mean and standard deviation, respectively). "$x, y - z$" represents $x$ and $y$ transfer to $z$, while "$x - z$" is transfer from $x$ to $z$.

ments, the **centipede** robotics have the most complex physical structures and CAT can solve these tasks well. Therefore, we have sufficient reasons to believe that CAT can achieve significant performance in other domains.

To prove this, we choose **Snake-x** as the additional environment, which has a similar structure to a snake. The goal of the agent is to move as fast as possible but the average rewards will eventually converge to around 400. We consider 400 as the solved score for SnakeSix and take the average required timesteps (M means one million steps) required for convergence as the evaluation criterion. We supplement the experiments with three source policies to verify that CAT can scale to more source tasks. Besides, we also add one source policy **Ant** with a completely different physical structure to verify the ability of CAT to avoid negative transfer.

Table 1 shows the performance of CAT and learning from scratch in our additional experiments, where "3,4,5-6" represents **SnakeThree**, **SnakeFour**, and **SnakeFive** transfer to **SnakeSix**. As the result shows, CAT can extract knowledge from more source policies. From experience, robots with more similar morphology can provide more knowledge, corresponding to robots with a similar number of joints in our experimental setting. CAT can provide accurate weights of each source policy based on the self-adaptation module when using more than two source tasks. Therefore, we only use two source policies for simplicity in our main experiments. Besides, CAT can also significantly improve learning efficiency even if there is one policy **Ant** which is expected

to provide negative transfer. This is because CAT has two mechanisms to avoid negative transfer. First, if a source policy that may cause negative transfer is added, it will get a particularly small weight, which will not affect the transfer performance. Second, CAT uses two weighted linear combinations. At the beginning of the training, the agent selects an action relying more on source policies to gain assistance but focuses more on the target task as the training continues. In this way, CAT can effectively avoid negative transfer.

Table 1: Performance of CAT and PPO on Snake-6, where 'M' denotes million training steps.

| Method | Time to Threshold | Rewards |
|---|---|---|
| PPO | 1.34M (±0.10) | 449.44 (±15.01) |
| CAT(3,4,5-6) | 0.70M (±0.09) | 452.10 (±17.12) |
| CAT(3,4,Ant-6) | 0.80M (±0.07) | 458.70 (±17.55) |

## 4.2 ABLATION STUDIES

To better illustrate the effectiveness of our proposed method, we analyze the contribution of the mutual information loss to verify the necessity of the correction module for the state embedding learning. Besides, we remove the correction module to see whether the CAT agent can achieve good performance only by relying on the self-adaptation module and state embeddings that can not satisfy properties (3) and (4). The ablation studies are designed as follows:

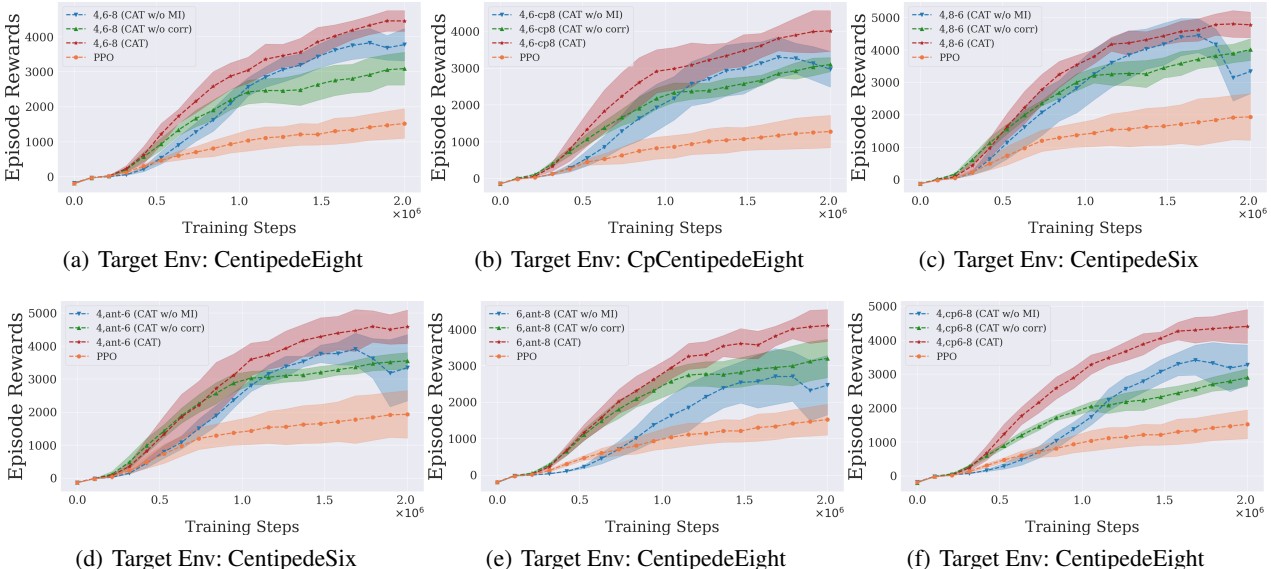

Figure 5: Ablation studies on the contribution of the mutual information loss and the correction module: *CAT w/o MI* and *CAT w/o corr*.

- *CAT w/o MI*: Update state encoders without $L_{\mathrm{MI}}$.
- *CAT w/o corr*: Update state encoders without $\{L_{\mathrm{cyc}}, L_{\mathrm{corr}}\}$.
- *CAT w/o corr and adapt*: Note that MIKT-MULTI can be seen as a version of CAT without the self-adaptation module and correction module, which we have shown in Figure 4.

Figure 5 shows the influence of these different parts on the performance of CAT-PPO. Before analyzing the experimental results, we note that it is usually harder to learn in the early stages of training for *MIKT w/o MI*, which removes the mutual information loss, as discussed earlier in Wan et al. [2020]. But we can see that *CAT w/o MI* still has very impressive performance compared to *CAT w/o corr* in most cases. This indicates it still achieves good transfer performance even without the mutual information loss, which confirms the effectiveness of our proposed properties and correction module. Besides, *CAT w/o corr* has a significant improvement compared to MIKT-MULTI, which confirms the effectiveness of the self-adaptation module. It is obvious that CAT is the most performant in all methods. This supports our view that the sufficiently trained state embeddings can indeed improve the transfer performance by satisfying the four properties at the same time.

Table 2 shows the average episode rewards without different modules in centipede 4,6-8 including the self-adaptation module. Note that for simplicity we do not analyze the contribution of the self-adaptation module separately in Figure 5, which can be verified by comparing the performance of *CAT w/o corr* and MIKT-MULTI. As the result shows, CAT is the most performant in all methods. All above re-

sults confirm that each component in CAT is necessary and important for effective and efficient transfer in DRL.

Table 2: Contributions of different modules of CAT in Centipede4,6-8.

| Method | Average Return |
|---|---|
| PPO | 1660.7 (±284.5) |
| MIKT (6-8) | 2940.0 (±357.0) |
| CAT | **4684.4 (±452.1)** |
| CAT w/o MI | 3972.1 (±312.0) |
| CAT w/o corr | 3097.8 (±289.8) |
| CAT w/o self-adapt | 3381.4 (±286.7) |
| MIKT-MULTI (w/o corr and self-adapt) | 2441.5 (±413.5) |

## 5 RELATED WORK

**Same-domain transfer and cross-domain transfer**
In same-domain transfer, one mainstream method to accelerate DRL is policy distillation, which is extended by Rusu et al. [2016]. Parisotto et al. [2016] mimics the behavior of source policies during the target policy learning process. However, this method highly relies on the task similarity, which restricts its generality. Schmitt et al. [2018] presents an auxiliary objective which distills knowledge from source policies by minimizing the cross-entropy loss between the source and target policy distributions over actions. However, this method uses an evolution strategy to adjust the hyperparameters which increases the computational complexity. Tao et al. [2021] propose to combine

multiple transfer manners, like policy distillation and value function reuse to facilitate more efficient DRL. However, they assume that the reward function of the target task is known, which is difficult to achieve in our problem setting. Successor features and generalized policy improvement also reuse source policies (value functions) directly in the target task [Barreto et al., 2017, 2019]. However, all these methods share the same limitation that cannot be applied to tasks with different state-action spaces which is more practical in real-world scenarios. Recently, Gupta et al. [2017] learns invariant state feature spaces and matches the distributions of optimal trajectories in the source task to transfer skills between different agents. However, they need paired data to train embedding functions which is very expensive in real-world problems. Zhang et al. [2021] learn the mapping between the state-action space to reuse the source policy directly, which may not achieve optimal performance on the target task. Our work is most relevant to Mutual Information Based Knowledge Transfer (MIKT) [Wan et al., 2020]. Although MIKT is a successful approach for cross-domain transfer, it is still faced the problem of insufficiently trained state embeddings and the limitation of being able to transfer only a single source policy. CAT firstly proposes the properties that state embeddings should satisfy at the same time and achieves adaptive knowledge transfer from multiple source policies with different state-action spaces.

**Domain Randomization and Domain Adaptation in RL**
Domain randomization aims to learn a policy with generalization capability which is trained on multiple source domains, hoping to perform well in the target domain [Tobin et al., 2017, Slaoui et al., 2019]. It focuses more on common features between domains by training on multiple source domains. However, this kind of method requires multiple source domains to be available for training, which is a strong assumption compared to the requirement that only pre-trained source policies are needed. Besides, domain randomization is very sensitive to changes in the number of domains, which greatly affects the complexity of training. Some domain adaptation works in RL use image-to-image translation to pair the pixel-based states in the source and target domain, but it has additional computational cost overhead for the image translation [Pan et al., 2017, Gamrian and Goldberg, 2019]. Other works focus on learning a common state representation to solve the problems mentioned above [Xing et al., 2021, Roy and Konidaris, 2021]. However, works in this field do not have a clear benchmark for the difference between the two domains and most of them focus on the problem of observation adaptation. While the source domain $\mathcal{D}_{\text{source}}$ and target domain $\mathcal{D}_{\text{target}}$ have different state space $\mathcal{S}$ (visual observations), the action space $\mathcal{A}$ and other properties should remain the same or have some similarity. The main difficulty in our work is how to achieve knowledge transfer among totally different MDPs, which is a more difficult task that these methods cannot be applied.

# 6   CONCLUSION AND FUTURE WORK

In this work, we firstly propose a novel framework called Cross-domain Adaptive Transfer (CAT) which adaptively transfers knowledge from multiple cross-domain policies. CAT is composed of three main components: the agent module, the self-adaptation module, and the correction module. Using the agent module and correction module, we firstly propose four properties that the learned state-action correspondence should satisfy. Then we design the corresponding optimization objectives to learn state and action embeddings to deal with the mismatch in the state-action space of source and target tasks. The self-adaptation module learns to decide when and which source policy is better to transfer by evaluating them on the target environment. The average performance is used to derive the weighting factors so that we can combine these different source policies. The agent module allows our agent to distill knowledge from source policies, select actions to execute in the target environment and learn a high-performing policy. Experimental results show that CAT significantly accelerates RL and outperforms other cross-domain transfer methods. In this paper, we use the average performance over a fixed number of episodes as the weight of each source policy in the next entire iteration. However, each source policy may only be helpful in a part of the state space. It's worthwhile investigating which source policy performs better in which region to facilitate fine-grained transfer. Another direction is to learn a unified embedding space for all source domains and the target domain to improve the generalizability of the method. Besides, leveraging prior human knowledge Zhang et al. [2020] or synthesizing white-box knowledge Cao et al. [2022] for a better transfer learning is worth further study.

## Acknowledgements

The work is supported by the National Natural Science Foundation of China (Grant Nos.: U1836214, 62106172), the new Generation of Artificial Intelligence Science and Technology Major Project of Tianjin under grant: 19ZXZNGX00010 and the Science and Technology on Information Systems Engineering Laboratory (Grant No. WDZC20205250407). Part of this work has taken place in the Intelligent Robot Learning (IRL) Lab at the University of Alberta, which is supported in part by research grants from Alberta Innovates; the Alberta Machine Intelligence Institute (Amii); a Canada CIFAR AI Chair, Amii; Compute Canada; Mitacs; and NSERC.

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
