# OpenReview forum: "Cross-domain Adaptive Transfer Reinforcement Learning Based on State-Action Correspondence"
_auai.org/UAI/2022/Conference — UAI 2022 Poster_

### Official Review · Reviewer_dHA6 · 2022-04-12

**Q2(1) Originality/Novelty:** 3
**Q2(2) Significance/Impact:** 3
**Q2(3) Correctness/Technical Quality:** 3
**Q2(6) Clarity Of Writing:** 4
**Q6 Overall Score:** 7
**Q8 Confidence In Your Score:** 4

**Q1 Summary And Contributions:**

This paper proposes Cross-domain Adaptive Transfer (CAT) for deep RL. CAT learns the state-action correspondence from source tasks to a target task via a correction module, and then adaptively chooses weights for how to transfer knowledge from multiple source task policies to the target policy via a self-adaptation module.  CAT can be combined with existing DRL algorithms, and empirical results show improved learning performance when combining CART with PPO.

**Q2 Assessment Of The Paper:**

More detailed information regarding each of these aspects is given below:

**Q2(4) Quality Of Experiments (Optional):**

3: Good: The experimental evaluation is adequate, and the results convincingly support the main claims.

**Q2(5) Reproducibility:**

3: Good: Key resources (e.g., proofs, code, data) are available and key details (e.g., proofs, experimental setup) are sufficiently well-described for competent researchers to confidently reproduce the main results.

**Q3 Main Strengths:**

* The desired properties for state embeddings provided in section 3.2 and the correction module that incorporates these properties when learning embeddings.
* The framework for the self-adaption module
* Good empirical results
* The paper is well written

**Q4 Main Weakness:**

* Reproducibility would be improved if source code is provided.
* Experiments are only carried out in one domain, and it would be be good to see how CAT performs in other domains.
* Only results for learning tasks with two source tasks are provided, and so it's not how clear how CAT may scale to more/many source tasks.



**Q5 Detailed Comments To The Authors:**

I think enabling transfer from multiple source policies with potentially different state and action spaces from the target policy is a difficult and worthwhile problem with potential impact on multiple areas of AI.  Overall an interesting paper with some novel ideas and good results.

It would be interesting to see how weights vary over time during learning.  Is it the case that the self-adaption module first weighs one source task higher, and then as learning progresses switches to weighting the other task higher (sort of like automated curriculum selection)?

I would like to see experimental results for learning tasks with more than two source tasks to evaluate scalability.  I'd also like to see an example with a source task that is expected to provide negative transfer to see if the self-adaption module filters out the negative transfer source task.



**Q7 Justification For Your Score:**

I think the paper provides some novel ideas and good results for the hard and meaningful problem of multiple source task transfer which is why I think it should be accepted.  The paper could be stronger, and I would probably give it a higher score, if the points I listed in the weaknesses and comments to the authors sections were addressed.

**Q9 Complying With Reviewing Instructions:**

1: Yes.

---

### Official Review · Reviewer_NoHD · 2022-04-14

**Q2(1) Originality/Novelty:** 2
**Q2(2) Significance/Impact:** 2
**Q2(3) Correctness/Technical Quality:** 3
**Q2(6) Clarity Of Writing:** 2
**Q6 Overall Score:** 4
**Q8 Confidence In Your Score:** 3

**Q1 Summary And Contributions:**

In this paper, a policy transfer method for reinforcement learning is proposed, assuming that the state and action spaces of the MDPs where the source policies are trained can be different from the ones in the target MDP. The proposed method includes three key modules learning the mapping between the source-target state-action spaces, the weights of the source policies, and the integration of the source and target policies. The proposed method is tested under continuous action control tasks.

**Q2 Assessment Of The Paper:**

More detailed information regarding each of these aspects is given below:

**Q2(4) Quality Of Experiments (Optional):**

2: Fair: The experimental evaluation is weak: important baselines are missing, or the results do not adequately support the main claims.

**Q2(5) Reproducibility:**

2: Fair: Key resources (e.g., proofs, code, data) are unavailable but key details (e.g., proof sketches, experimental setup) are sufficiently well-described for an expert to confidently reproduce the main results.

**Q3 Main Strengths:**

1. The paper studies a meaningful problem: studying policy transfer across heterogeneous state-action spaces is a very meaningful topic.



**Q4 Main Weakness:**

1. The paper lacks clear descriptions of the learning scenario, in special the requirements for the proposed method. For example, in algorithm 1, the source buffer is included. How are the samples in the buffer are collected? Is the target learner allowed to access the source MDP for collecting these samples? There is no clear description in the paper.

2. The experimental part is somehow insufficient. As mentioned in the introduction, there are several methods proposed recently to study the heterogeneous state-action space policy transfer. While only one of them is included in the experiments as the comparison baseline.

**Q5 Detailed Comments To The Authors:**

1. As mentioned in Q4, clearer descriptions of the learning scenario are necessary to be included. I think how the source buffer can be generated is essential for the task. If the target learner is allowed to access the source MDP, the hardness of the problem is significantly reduced. On the other hand, if the source buffer is collected beforehand independently to target learning, then how can we guarantee that good feature space mapping can be learned when no paired source-target samples exist?

2. The detailed experimental setups are missing in the paper. Some important information should be described. For example, how the source and target state-action spaces are set? How the source policies are obtained? How good are they?

3. As mentioned in Q4, more comparison baselines are needed to be included in the experiments.

Minor:
In Fig. 3 and 4, the overlaps between the legends and the results need to be avoided.

**Q7 Justification For Your Score:**

To summarize, even though I think the paper studies an interesting and meaningful problem, it has non-ignorable issues in learning setup description and experimental evaluation.

**Q9 Complying With Reviewing Instructions:**

1: Yes.

---

### Official Review · Reviewer_uJmW · 2022-04-20

**Q2(1) Originality/Novelty:** 2
**Q2(2) Significance/Impact:** 2
**Q2(3) Correctness/Technical Quality:** 3
**Q2(6) Clarity Of Writing:** 4
**Q6 Overall Score:** 6
**Q8 Confidence In Your Score:** 2

**Q1 Summary And Contributions:**

Existing TL methods to improve RL efficiency usually assume the same state-action spaces across tasks and only transfer from a single source policy. To handle these challenges, the authors proposed a cross-domain adaptive transfer deep reinforcement learning method (CAT), which considered different state-action spaces across tasks and multiple source policies. Experiments show the efficacy of their method on multiple continuous action control tasks.

**Q2 Assessment Of The Paper:**

More detailed information regarding each of these aspects is given below:

**Q2(4) Quality Of Experiments (Optional):**

2: Fair: The experimental evaluation is weak: important baselines are missing, or the results do not adequately support the main claims.

**Q2(5) Reproducibility:**

1: Poor: Key details (e.g., proof sketches, experimental setup) are incomplete/unclear, or key resources (e.g., proofs, code, data) are unavailable.

**Q3 Main Strengths:**

The authors proposed a transfer framework, CAT, consisting of three main components: an agent module, a self-adaptive
module, and a correction module, to solve the problem from different state-action spaces and multiple source policies.
The paper is well written and the work looks interesting. Experiments settings are reasonable and the results show the efficacy of their method.

**Q4 Main Weakness:**

I have some concerns below:

1. It is not very clear whether the proposed method could achieve optimal performance since it has no theoretical guarantees despite the empirical performance.

2. In the ablation study, the authors only verify the necessity of the correction module of the method but seem to ignore the self-adaption module. It might be better to explain why. By the way, the self-adaptation module made use of the total return $u_i$ of each source policy on the target environment, which implies that such a module employed the information from the target domain. Hence, it seems that this module matters much in the proposed method and it is worthwhile to do the ablation study towards the self-adaptation module.

3. This paper lacks the code. It might be better to make the code available so that the experimental results could be reproducible.

**Q5 Detailed Comments To The Authors:**

Please see Q3-Q4.

**Q7 Justification For Your Score:**

If the authors could address the concerns listed in Q4, I think it is worthwhile to raise the score.

**Q9 Complying With Reviewing Instructions:**

1: Yes.

---

### Decision · Program_Chairs · 2022-05-15

**Decision:**

Accept (Poster)

**Comment:**

Meta Review: This paper proposes a transfer learning method for Deep reinforcement learning.  While previous work mainly considers TL across tasks with the same state-action spaces, this paper investigates transferring across domains with different state-action spaces.

The reviewers have divergent views on the paper, and the disagreements were not resolved after extensive discussions. Here are the concluding remarks of the reviewers at the end of the discussions:

uJmW: Thanks for the authors' clarification, which addresses my concerns. I will increase the score to 6.

dHA6 (overall score: 7): After reading over all the other reviews, as well as very good exchanges between the authors and reviewers, my opinion and score for the paper hasn't really changed. The problem being tackled in the paper is both difficult and significant, and although the authors may not have theoretical guarantees or a completely general solution to all problems of this nature (should one even exist), I think the contributions of the paper are still strong enough to meet the bar for acceptance and would be useful to others working on similar problems.

NoHD (overall score: 4):  have some experience in studying policy transfer between two MDPs with different state/action spaces. In my view, this problem is very hard when no data with views in both MDPs exists. Thus I am quite curious about how the paper addresses this problem. I am somehow disappointed to see that this is not addressed by specified techniques but just by some implicit assumptions. Overall, I appreciate the novel setting studied in the paper. I also think the proposed method would work effectively in some situations. But in my view, the method is more a heuristic one than a sound approach that works under the general situation. I would keep my score, but I would not be upset if other reviewers think the paper is worth an acceptance.

I recommend accept because the problem setting is novel and significant, and the paper might inspire further research on the topic.